# Field evaluation of the effect of *Aspergillus niger* on lettuce growth using conventional measurements and a high-throughput phenotyping method based on aerial images

Patrick Vieira Silva[1], Lucas Medeiros Pereira[1], Gustavo de Souza Marques Mundim[1], Gabriel Mascarenhas Maciel[1], Rodrigo Bezerra de Araújo Gallis[2], Gilberto de Oliveira Mendes[1]*

1 Instituto de Ciências Agrárias, Universidade Federal de Uberlândia, Monte Carmelo, Minas Gerais, Brazil,
2 Instituto de Geografia, Universidade Federal de Uberlândia, Monte Carmelo, Minas Gerais, Brazil

* gilbertomendes@ufu.br

## Abstract

Plant microbiome engineering is a promising tool to unlock crop productivity potential and exceed the yield obtained with conventional chemical inputs. We studied the effect of *Aspergillus niger* inoculation on in-field lettuce (*Lactuca sativa*) growth in soils with limiting and non-limiting P concentrations. Lettuce plants originating from inoculated seeds showed increased plant diameter (6.9%), number of leaves (8.1%), fresh weight (23.9%), and chlorophyll content (3.8%) as compared to non-inoculated ones. Inoculation of the seedling substrate just before transplanting was equally efficient to seed inoculation, while application of a granular formulation at transplanting did not perform well. Plant response to P addition was observed only up to 150 kg $P_2O_5$ ha$^{-1}$, but *A. niger* inoculation allowed further increments in all vegetative parameters. We also employed a high-throughput phenotyping method based on aerial images, which allowed us to detect changes in plants due to *A. niger* inoculation. The visible atmospherically resistant index (VARI) produced an accurate prediction model for chlorophyll content, suggesting this method might be used to large-scale surveys of croplands inoculated with beneficial microorganisms. Our findings demonstrate that *A. niger* inoculation surpasses the yield obtained with conventional chemical inputs, allowing productivity gains not reached by just increasing P doses.

## Introduction

World population is forecast to reach 9.7 billion by 2050, which will demand doubling food production [1]. The sustainable way to increase food production is by raising crop yields on existing croplands, minimizing land clearance and greenhouse gas emissions [2, 3]. To reach a crop's yield potential, a perfect combination of favourable climate, pest and disease control, and non-limiting nutrients and water is necessary. However, some estimations suggest that average crop yields plateau at 75–85% of the crop yield potential [4]. This gap between farm

**Data Availability Statement:** The datasets generated during the current study are available in

the Mendeley Data repository, DOI: 10.17632/82c548svpy.1.

**Funding:** This work was supported by the Fundação de Amparo à Pesquisa do Estado de Minas Gerais (FAPEMIG), grant number APQ-01842-17 to GOM, the Conselho Nacional de Desenvolvimento Científico e Tecnológico (CNPq), grant number 407793/2021-6 to GOM, and NOOA Ciência e Tecnologia Agrícola Ltda, grant to GOM and PVS. The funders had no role in study design, data collection and analysis, decision to publish, or preparation of the manuscript.

**Competing interests:** : I have read the journal's policy and the authors of this manuscript have the following competing interests: a) Research support by the company NOOA Ciência e Tecnologia Agrícola Ltda to PVS and GOM, although the company had no role in the conceptualization, design, data collection, analysis, decision to publish, or preparation of the manuscript; b) Patent application by the Universidade Federal de Uberlândia, Fundação de Amparo à Pesquisa do Estado de Minas Gerais, and NOOA Ciência e Tecnologia Agrícola Ltda; inventors GOM et al.; application number BR 10 2020 000628 2, status: under analysis; specific aspect of manuscript covered in patent application: granular formulation of A. niger used in experiments. All other authors do not have any conflict of interest to declare. This does not alter our adherence to PLOS ONE policies on sharing data and materials.

yields and the crop yield potential is attributed to the difficulty for farmers to achieve the perfect management and to the low response to additional inputs when the yield approaches the plateau, which discourage additional investments [4].

A commonly overlooked component of crop yield potential is the plant microbiota, which plays an important role in plant growth, health, and resilience to stress [5]. The benefits of plant microbiota for agriculture have gained substantial attention recently. Many studies reported diverse plant beneficial microorganisms (PBM) showing multiple mechanisms to promote plant growth and health, such as phytohormone production, facilitating nutrient acquisition, enhancing tolerance to salinity and drought, and disease suppression [5–9]. However, plant domestication may have undermined beneficial microbial associations in modern crop cultivars [10]. In this scenario, plant microbiome engineering has been proposed as a way to reinstate beneficial plant-microorganism associations in agroecosystems [10, 11]. Therefore, inoculation of PBM can raise crop yield and diminish the gap in the crop yield potential.

*Aspergillus niger* v. Tiegh is a PBM that has multiple mechanisms of plant growth promotion, such as phytohormone production [12, 13] and solubilization of P and K [14–16]. Inoculation of *A. niger* enhanced the growth of coffee (*Coffea arabica* L.) [17], maize (*Zea mays* L.) [13], and vegetable seedlings [18]. Moreover, *A. niger* can release P fixed to the soil [15] and, thus, may contribute to improve crop P use efficiency and access the soil legacy P, alleviating the pressure on world phosphate reserves [19]. We hypothesized that *A. niger* could enhance lettuce (*Lactuca sativa* L.) yield in soils with limiting and non-limiting concentrations of P. Lettuce is a worldwide distributed crop [20] in which inoculation can reinstate important microbial associations that may improve growth and yield [21, 22]. Thus, the aim of this research was to evaluate lettuce growth under different methods of *A. niger* inoculation and P availability levels.

## Materials and methods

### Experimental site

The experiment was carried out from April to July 2021 at the Vegetable Experimental Station of the Federal University of Uberlândia, located in Monte Carmelo, Minas Gerais state, Brazil (18˚42'43.19" S, 47˚29'55.8" W, 873 m altitude). Climate of the region is characterized according to Köppen as Aw-tropical, with hot and humid summer and cold and dry winter. Average daily temperature during the experiment period was 21.1˚C, with an average minimum of 14.9 and a maximum of 27.3˚C. Average relative humidity was 65.6%, with an average minimum of 44.9 and a maximum of 86.4%.

The soil presented clay texture (sand 25.5%, silt 10%, and clay 64.5%), pH (in $H_2O$) 5.1, 23.8 mg P $dm^{-3}$ (extracted by Mehlich-1), 113.4 mg K $dm^{-3}$, 352.9 mg Ca $dm^{-3}$, 40.1 mg Mg $dm^{-3}$, 24.3 mg $Al^{3+}$ $dm^{-3}$, and organic matter 12 g $kg^{-1}$.

### Experimental setup

The experiment was arranged in a randomized block design in a 4 x 3 factorial scheme. The first factor consisted of four inoculation methods: seed inoculation (SI) at sowing, conidial suspension (CS) applied to seedling substrate just before transplanting, granular formulation (GR) applied to the planting hole at transplanting, and a non-inoculated (NI) control. All inoculated treatments received $10^2$ conidia $plant^{-1}$ (see details in the section Production and application of *Aspergillus niger* inoculants). The second factor consisted of three P doses: 0, 150, and 300 kg $P_2O_5$ $ha^{-1}$. P was supplied as triple superphosphate (18.9% P). The experiment was carried out with four repetitions, adding up to 48 plots. Each plot consisted of 16 plants and the four central plants were considered for evaluation.

Lettuce seedlings were produced in polystyrene trays with 128 cells (40 cm$^3$ cell$^{-1}$) filled with a commercial coconut fiber substrate (Technes, São Paulo, SP, Brazil). Substrate presented pH 5.6, humidity 48%, density 260 kg m$^3$, 0.8% fertilizer, and 0.2% corrective. Lettuce seeds of the UFU-197#2#1#1 genotype were obtained from the Biofortified Lettuce Genetic Improvement Program/UFU. Two seeds were sowed in each cell and after 10 days seedlings were thinned to one per cell. Seedlings were fertigated weekly with 13.4 mL per cell of a solution containing (total amount per cell): 5.02 µg N, 2.19 µg P, 5.55 µg K, 0.83 µg Mg, 0.083 µg Zn, 0.025 µg B, 0.0083 µg Fe, 0.083 µg Mn, and 0.92 µg S. Irrigation was performed daily according to plant needs. Trays were placed on 70-cm high benches in a greenhouse (5 x 6 m, height 3.5 m) covered with 150-µm clear plastic film and side curtains of anti-aphid white screen.

Seedlings were transplanted to beds 48 days after sowing. Each experimental plot had 16 plants distributed in four rows with four plants each spaced 0.25 m apart. Spacing between plots and beds was 0.5 m. Nitrogen and potassium were supplied at 50 kg K ha$^{-1}$ and 150 kg N ha$^{-1}$ as urea and KCl, respectively [23]. Foliar applications of B, Ca, and K were carried out weekly using a commercial fertilizer formulation (3.8% B, 19% Ca, and 1% K$_2$O) at 100 g ha$^{-1}$. Irrigation was carried out daily. Cultural traits were performed according to plant needs.

## Production and application of *Aspergillus niger* inoculants

The fungus *A. niger* FS1 was previously isolated from soil under native forest in Viçosa, Minas Gerais, Brazil [16]. Fungal conidia were produced in Petri dishes containing potato dextrose agar (PDA, Sigma-Aldrich, Saint Louis, MO, USA) incubated at 30˚C for 10 days. Conidia were collected in a sterile Tween 80 0.01% (v/v) solution. The conidial suspension was vacuum filtered through a membrane with 0.45 µm pores and conidia retained on the membrane were dried in a desiccator with silica gel at room temperature (25˚C) for 24 h. Dry conidia were collected and stored in microtubes at room temperature [17]. Dry conidia mass contained 4.5 x 10$^7$ conidia mg$^{-1}$, as counted in a Neubauer chamber.

Conidia were inoculated to plants at a concentration of 10$^2$ conidia plant$^{-1}$ [18] in three ways: seed inoculation (SI), application of a conidial suspension to seedling substrate just before transplanting (CS), and granular formulation applied to the planting hole at transplanting (GR).

## Seed inoculation

Dry conidia of *A. niger* FS1 were suspended in sterile water at 0.005 mg conidia mL$^{-1}$. To deliver 10$^2$ conidia per seed, 460 lettuce seeds were covered with 0.69 mL of this suspension two hours before sowing via pipetting and homogenized manually. Seeds were kept moist until sowing in trays.

## Seedling inoculation with conidial suspension

Dry conidia of *A. niger* FS1 were suspended in sterile water at 0.002 mg conidia L$^{-1}$. Each seedling received 3 mL of this conidial suspension 30 min before transplanting. The conidial suspension was added individually in each tray cell using a sterile micropipette.

## Granular formulation applied to the planting hole at transplanting

The granular formulation was produced by mixing dry conidia with 26.5 g wheat flour, 3.8 g corn starch, 2.25 g granulated sugar, and 15 mL deionized water [17]. The amount of dry conidia added to the mixture was 0.013 mg to yield a concentration of 10$^2$ conidia per granule

(average mass of each granule was 22.6 mg). The mixture was extruded into filaments of 2 mm diameter, which were cut to 2 mm in length and dried in an oven with forced air circulation at 50˚C for 48 h [17]. The resulting granules had a cylindrical form with approximate 2 mm in height and a 2 mm diameter base. At transplanting, one granule was added to the planting hole just below the seedling root.

## Analytical procedures

**Vegetative growth parameters.**   Measurements of the plant diameter, stem diameter, fresh weight, number of commercial leaves (longer than 5 cm in length), and SPAD index were taken from the four central lettuce plants in each plot. The SPAD index was measured with a chlorophyl meter (Minolta SPAD-502CFL1030) in three leaves of each plant, taking reads from two opposed lateral leaves and from an intermediary one. All measurements were done at 33 days after transplanting for plant diameter and SPAD index and at day 34 for the rest of the variables.

## Image-based phenotyping

Aerial images of the lettuce canopy were taken 33 days after transplanting by a remotely piloted aircraft (Phantom 4 Pro drone) equipped with a RGB camera. The flight and photo-graphing were controlled by the DJI and DroneDeploy software and performed at a height of 20 m, longitudinal and lateral overlap of 80%, direction at -93˚, and speed of 3 m s$^{-1}$. The images obtained were processed with the Pix4Dmapper software to generate an orthoimage with 1 cm of GSD (Ground Sample Distance). A sample of the orthoimage is shown in Fig 1 and the complete image is available as Supplementary Data (S1 Fig).

The orthoimage was processed using the package *FIELDimageR* for R [24]. The histogram of the HUE vegetation index was generated and used to determine a cut-off value of 1.5 to segment the vegetation from the soil background. After segmentation, histograms of vegetation indices related to RGB bands were generated and vegetation indices were calculated according to equations shown in Table 1.

A Pearson's correlation matrix between the vegetation indices and the vegetative growth parameters was generated using the package *Hmisc* for R. Correlation coefficients were tested at $p = 0.05$. The vegetation index with the highest correlation coefficients with the measured vegetative growth parameters was chosen for fitting prediction models using the machine learning software Waikato Environment for Knowledge Analysis (WEKA, version 3.9.5).

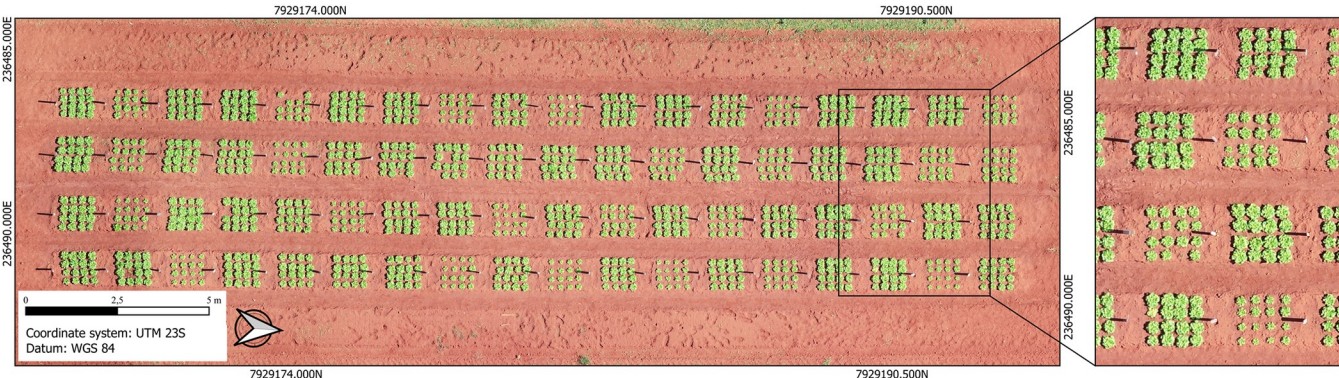

**Fig 1. Orthoimage showing the aerial view of the experiment.** The zoomed image on the right highlights the spatial resolution of the orthoimage generated with a GSD of 1 cm.

**Table 1. Equations for calculation of vegetation indices from red (R), green (G), and blue (B) bands of images.**

| Vegetation index | Equation | Reference |
|---|---|---|
| Overall Hue Index (HUE) | $arctan(2^*(B-G-R)/30.5^*(G-R))$ | [25] |
| Normalized Green Red Difference Index (NGRDI) | $(G-R)(G+R)$ | [26] |
| Green Leaf Index (GLI) | $(2G-R-B)/(2G+R+B)$ | [27] |
| Soil Color Index (SCI) | $(R-G)/(R+G)$ | [28] |
| Brightness Index (BI) | $\sqrt{((R^2+G^2+B^2)/3)}$ | [29] |
| Visible Atmospherically Resistant Index (VARI) | $(G-R)/(G+R-B)$ | [30] |
| Excess Green Index (ExG) | $2G-R-B$ | [31] |
| Modified Green Red Vegetation Index (MGVRI) | $(G^2-R^2)/(G^2+R^2)$ | [32] |

Linear regression models were fit using the percentage split option, where 90% of the dataset was considered for model training. The accuracy of fitted models was checked by the root mean squared error (RMSE) (Eq 1), and the normalized RMSE (NRMSE) (Eq 2).

$$RMSE = \sqrt{\frac{\sum_{i=1}^{n}(\hat{y}_i - y_i)^2}{n}} \qquad \text{Eq1}$$

$$NRMSE\ (\%) = \frac{RMSE}{\bar{y}} \times 100 \qquad \text{Eq2}$$

where:

$\hat{y}$: estimated value

$y$: observed value

$n$: number of samples

## Statistical analyses

Data from vegetative growth parameters were subjected to ANOVA and treatment means were compared by LSD test ($p = 0.05$) using the package *ExpDes.pt* for R [33]. Data were tested ($p = 0.05$) for normality and homoscedasticity by Shapiro-Wilk and O'Neil tests, respectively.

Multivariate analyses based on all vegetative growth parameters were performed by clustering and principal component methods. A dendrogram was constructed by the unweighted pair-group method with arithmetic mean (UPGMA) based on Euclidean distance. Validation of clustering was performed based on the cophenetic correlation coefficient [34] calculated in the software Genes [35]. The relative importance of variables on the dissimilarity between treatments was calculated as proposed by Singh [36]. Principal component analysis (PCA) was carried out based on the correlation matrix using the package *stats* for R.

## Results

### Vegetative growth parameters

Lettuce vegetative growth parameters were affected by *A. niger* inoculation and P dose, but these factors did not interact significantly ($p > 0.05$). Seed inoculation was superior to non-inoculated treatment for all vegetative growth parameters, except for stem diameter, increasing average plant diameter, number of leaves, fresh weight, and chlorophyll content (estimated by the SPAD index) by 6.9, 8.1, 23.9, and 3.8%, respectively (Fig 2A–2E). The inoculation with a conidial suspension applied to the seedling substrate just before transplanting produced similar results to seed inoculation for all vegetative growth parameters. On the other hand, plants

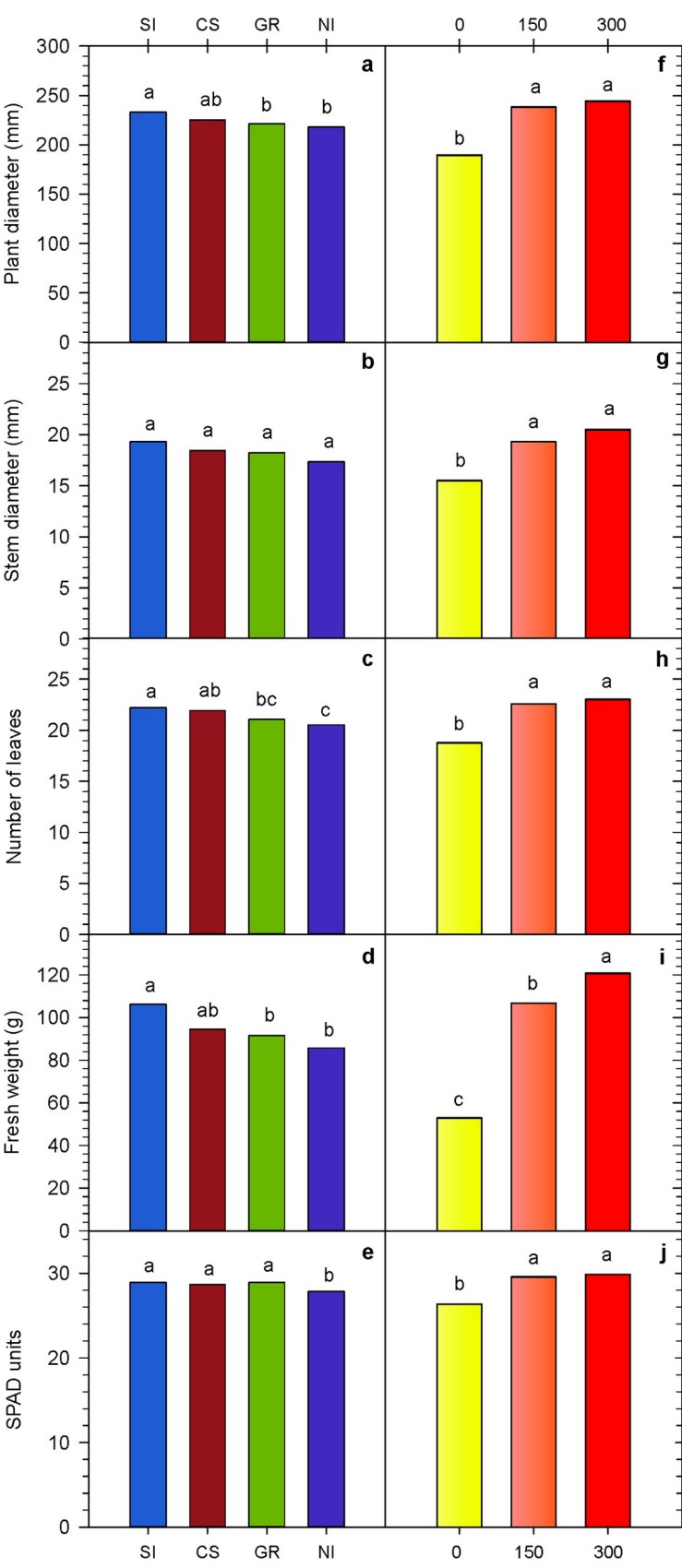

**Fig 2. Vegetative growth parameters of lettuce as affected by *Aspergillus niger* FS1 inoculation (a-e) and P dose (f-j).** Interaction between factors inoculation and P dose was not significant ($p > 0.05$). For each factor, treatments labelled with distinct letters are significantly different (LSD test, $p < 0.05$). SI: Seed inoculation, CS: Conidial suspension, GR: Granular formulation, NI: Non-inoculated. P doses: 0, 150, and 300 kg $P_2O_5$ ha$^{-1}$.

inoculated with the granular formulation grew like non-inoculated ones, only showing higher values of the SPAD index (Fig 2E).

P fertilizer application improved all vegetative growth parameters (Fig 2F–2J). However, plants fertilized with 150 and 300 kg $P_2O_5$ ha$^{-1}$ grew similarly, except for fresh weight, in which plants fertilized with 300 kg $P_2O_5$ ha$^{-1}$ showed higher values (Fig 2I).

Cluster analysis by UPGMA divided the combinations of inoculation methods and P doses in four clusters (Fig 3) at a level of 15% of dissimilarity [37]. Cluster I included treatments without P addition, cluster II was formed by the combination of seed inoculation and 300 kg $P_2O_5$ ha$^{-1}$, cluster III included non-inoculated treatments that received 150 and 300 kg $P_2O_5$ ha$^{-1}$, and cluster IV included the rest of inoculated treatments fertilized with 150 or 300 kg $P_2O_5$ ha$^{-1}$. Clusters formed by UPGMA had a cophenetic correlation coefficient of 0.874 (*t* test, $p < 0.01$), indicating that the dendrogram represents faithfully the matrix data. The SPAD index and fresh weight contributed the most for dissimilarity between treatments, representing 40.9 and 36.5% of the variability, respectively (Table 2).

Principal component analysis showed that most of the variance (96.5%) was explained by component 1 (Fig 4), reflecting the high correlation between the plant variables. All variables showed positive loadings for this component so that the higher the score of the treatment the higher the lettuce growth parameters. Scores allowed to group the treatments in the same clusters obtained by UPGMA. The highest score was presented by the combination seed inoculation + 300 kg $P_2O_5$ ha$^{-1}$ (2.9), followed by a group of treatments formed by inoculated plants fertilized with 150 or 300 kg $P_2O_5$ ha$^{-1}$, which presented an average score of 1.6. Non-inoculated treatments that received 150 and 300 kg $P_2O_5$ ha$^{-1}$ of the P dose presented a mean score of 0.3. Treatments without P addition presented the lowest scores, averaging -2.9.

## Image-based phenotyping of lettuce

Vegetation indices obtained from aerial images of lettuce canopy showed significant correlation ($p < 0.05$) with vegetative growth parameters measured in lettuce plants (Fig 5). Positive correlations with vegetative parameters were observed for VARI, NGRDI, GLI, ExG, and MGVRI since these indices are related to chlorophyll content. Conversely, negative correlations were obtained for BI and SI, which are related to soil parameters. The highest positive correlations were observed for VARI and, thus, this index was chosen for fitting models to predict vegetative growth parameters.

The model for predicting SPAD index was the most accurate (NRMSE 4.87%), followed by models for number of leaves, plant diameter, and stem diameter (Table 3). On the other hand, fresh weight model presented high prediction error (RMSE 32.75%).

## Discussion

Microbial inoculants are emerging as pivotal tools to unlock crop productivity potential and exceed the yield obtained with conventional chemical inputs. In this research, we demonstrate that lettuce plants inoculated with *A. niger* FS1 grew more than uninoculated ones even when plants had no P limitation. While the response to P addition was observed only up to 150 kg $P_2O_5$ ha$^{-1}$, *A. niger* inoculation allowed further increments in all vegetative parameters. Microbial plant growth promotion is a multifactorial process resulting from different mechanisms

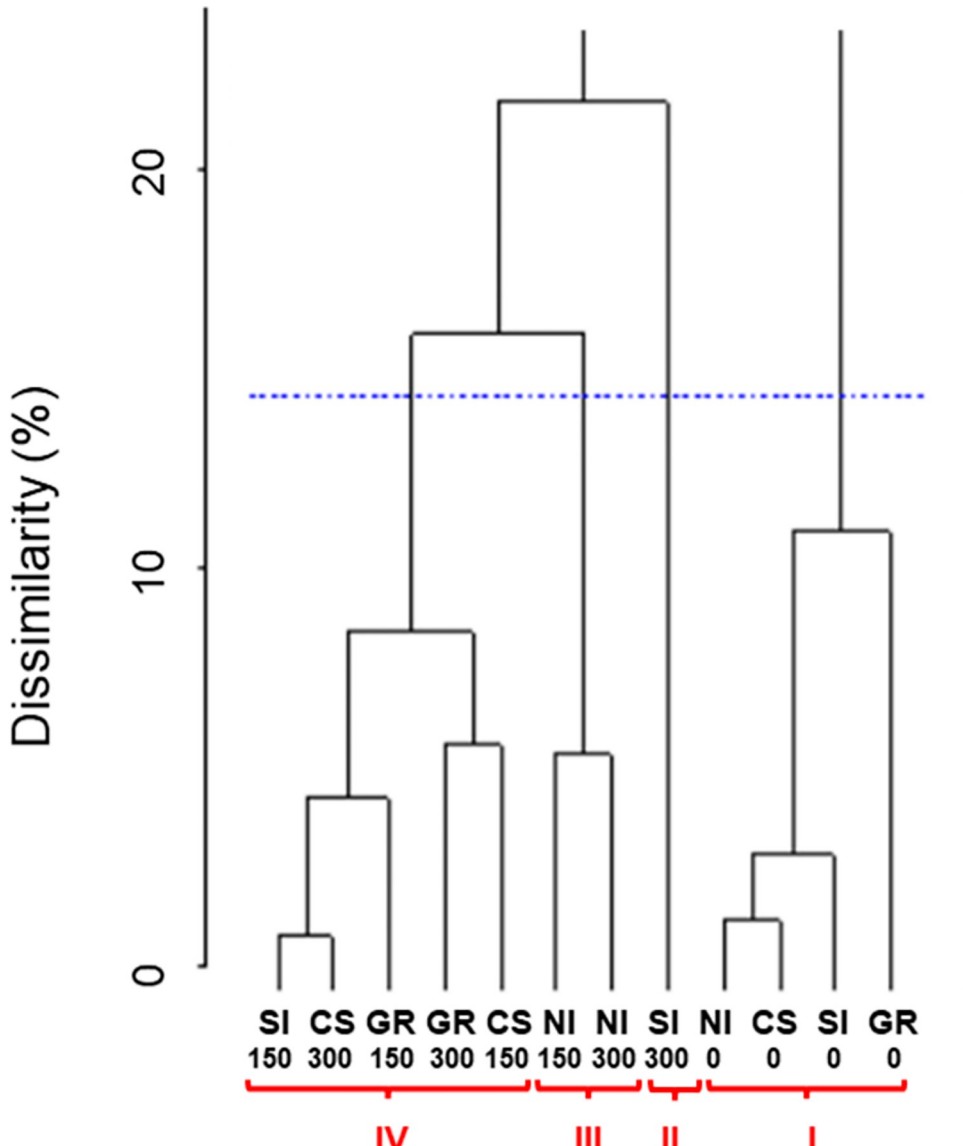

**Fig 3. Clustering of combinations of *Aspergillus niger* FS1 inoculation methods and P doses obtained by the unweighted pair-group method with arithmetic mean (UPGMA) based on Euclidean distance.** SI: Seed inoculation, CS: Conidial suspension, GR: Granular formulation, NI: Non-inoculated. P doses: 0, 150, and 300 kg $P_2O_5$ ha$^{-1}$.

**Table 2. Relative contribution of response variables for the dissimilarity between treatments calculated from Euclidean distance.**

| Response variable | Relative contribution (%) |
|---|---|
| Plant diameter | 10.94 |
| SPAD index | 40.97 |
| Stem diameter | 7.64 |
| Fresh weight | 36.56 |
| Number of leaves | 3.87 |

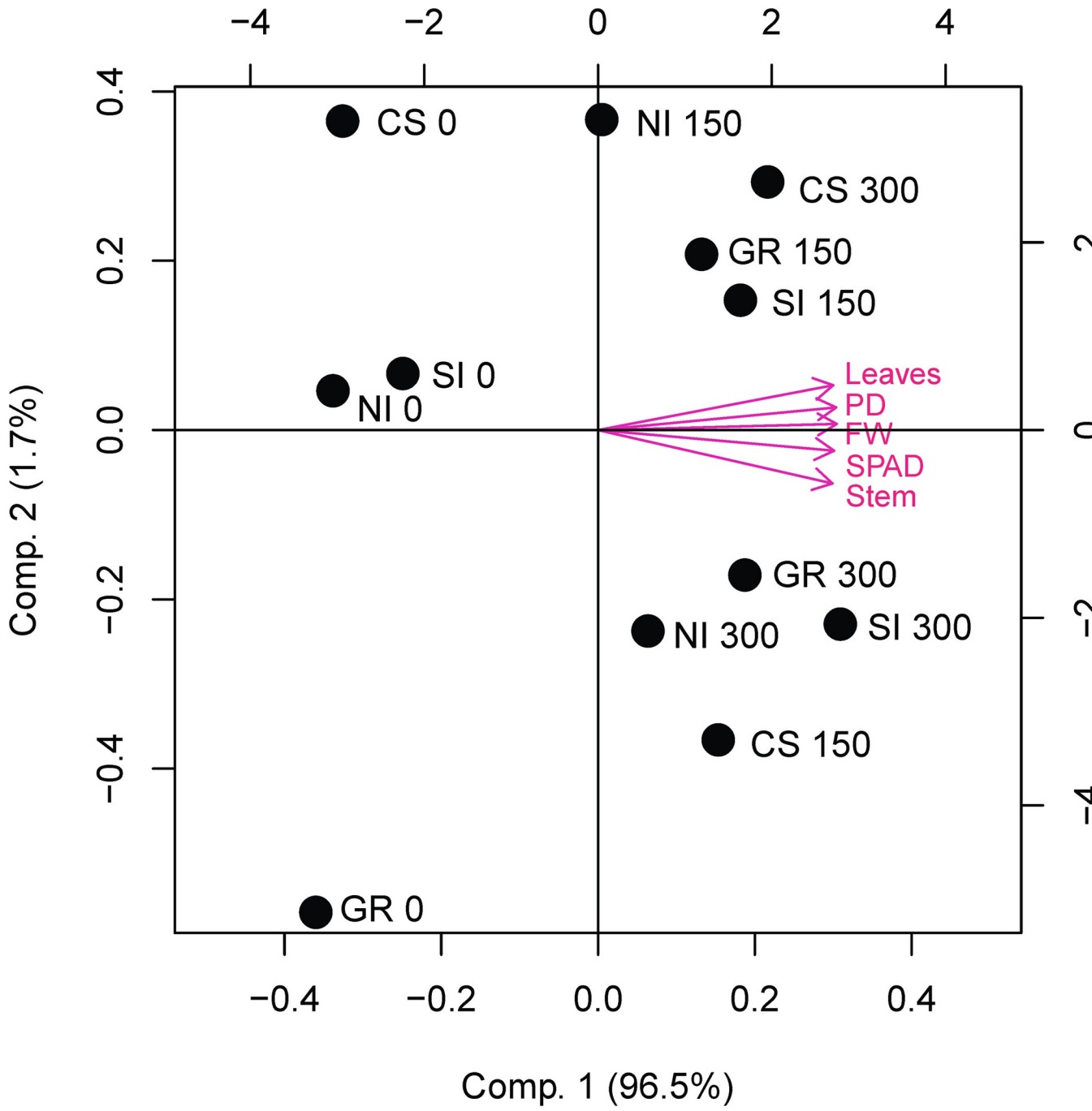

**Fig 4. Principal component analysis of combinations of *Aspergillus niger* FS1 inoculation methods and P doses.** Circles represent scores of treatments and arrows represent the loadings of plant characteristics on the components. PD: Plant diameter, FW: Fresh weight, SI: Seed inoculation, CS: Conidial suspension, GR: Granular formulation, NI: Non-inoculated. P doses: 0, 150, and 300 kg $P_2O_5$ ha$^{-1}$.

that enhance plant fitness under diverse environmental conditions [9, 38]. Moreover, different microbial mechanisms can be recruited by plants according to their needs. Plants cultivated in poor nutrient soils recruited preferentially phosphate solubilizers while microorganisms capable of phytohormone production predominated in rich nutrient soils [39]. *Aspergillus niger* strains have shown the capacity to produce phytohormones such as indoleacetic acid and

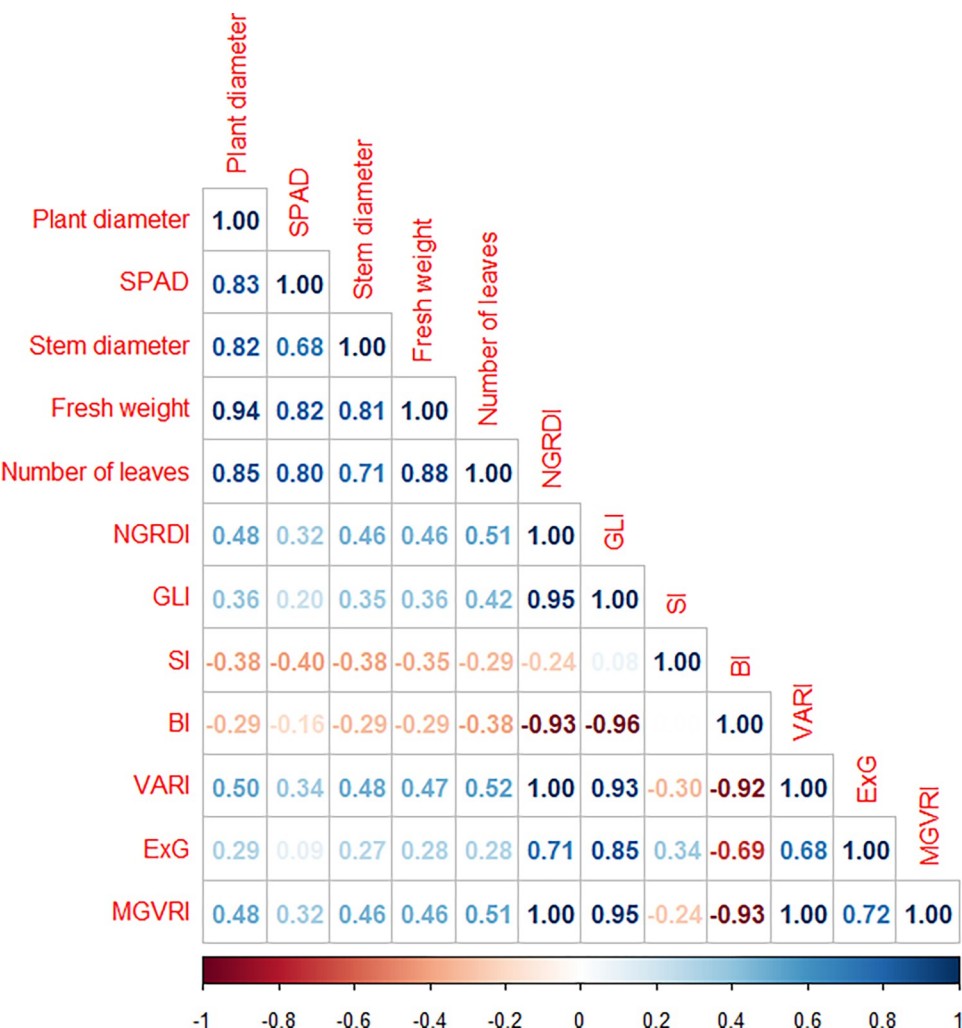

**Fig 5. Pearson's correlation matrix between plant growth parameters and vegetation indices.** Positive and negative correlations are shown in blue and red tones, respectively. NGRDI: Normalized Green Red Difference Index, GLI: Green Leaf Index, SI: Soil Color Index, BI: Brightness Index, VARI: Visible Atmospherically Resistant Index, ExG: Excess Green Index (ExG), and MGVRI: Modified Green Red Vegetation Index.

gibberellin [12, 13] and solubilize P and K from minerals [14–16]. Therefore, under nutrient scarcity, this species might release organic acids to make nutrients like P available to plants or, on the other hand, produce phytohormones to stimulate root and shoot development when

**Table 3. Regression models for prediction of lettuce vegetative growth parameters based on the VARI (visible atmospherically resistant index) vegetation index obtained from aerial images of the lettuce canopy.**

| Vegetative growth parameter | Equation | RMSE | NRMSE (%) |
|---|---|---|---|
| Plant diameter | 685.2348 VARI + 128.4624 | 21.68 | 9.67 |
| SPAD index | 31.6518 VARI + 24.1401 | 1.39 | 4.87 |
| Stem diameter | 72.6749 VARI + 8.1902 | 2.05 | 11.17 |
| Fresh weight | 763.8214 VARI—12.281 | 32.75 | 34.64 |
| Number of leaves | 57.0566 VARI + 13.4589 | 1.95 | 9.10 |

RMSE: Root-mean-square error, NRMSE: Normalized root-mean-square error

nutrients are not limiting plant growth [40]. This would allow the fungus to switch mechanisms according to the plant's needs, representing a smart choice for inoculant development. Our results indicate that well-nourished lettuce plants still benefit from inoculation with *A. niger*, suggesting that metabolic alterations due to fungal metabolites like phytohormones were taking place in plants. Indeed, we observed an increased chlorophyll content (SPAD index) in inoculated plants, which could be linked [41] with these fungal-promoted metabolic alterations.

The best ways to deliver *A. niger* conidia to plants were seed inoculation and application of a conidial suspension to the seedling substrate before transplanting. Both methods can be carried out in nurseries, allowing the introduction of *A. niger* as a core microorganism that may directly benefit the plant after transplanting to the soil as well as help to establish a network with beneficial indigenous microorganisms [11]. Moreover, a preliminary study showed that lettuce seedlings emerged from seeds inoculated with *A. niger* grew more and presented a well-developed root than uninoculated ones [18]. Additionally, once established in the seedling rhizosphere, *A. niger* would have a competitive advantage over native microbiota in the field, which probably was the advantage of these inoculation methods over in-furrow application of the granular formulation.

Although there is evidence that *A. niger* can release P fixed to soil [15], this mechanism was not observed in this study. The factors *A. niger* inoculation and P dose did not interact, demonstrating that the effect of *A. niger* on lettuce growth is independent of the P availability in the soil. The soil used in the experiment had 23.8 mg extractable P dm$^{-3}$. This amount was not enough to meet the plant's requirements once the addition of phosphate fertilizer increased plant growth. At this limiting P concentration, *A. niger* inoculation did not improve plant growth as would be expected. Microbial phosphate solubilization in the soil is probably lower than *in vitro* conditions since carbon and energy sources in the rhizosphere are disputed resources by the microbiota. This hampers the production of high concentrations of organic acids necessary for phosphate solubilization [16, 42, 43].

We used a high-throughput phenotyping method based on aerial images to evaluate changes in plants due to *A. niger* inoculation. The visible atmospherically resistant index (VARI) produced good prediction models for vegetative growth parameters, especially the SPAD index. The VARI is a vegetation index for quantitative estimation of vegetation fraction with the visible range of the spectrum [30]. Therefore, leaf pigments like chlorophyll and carotenoids have a major influence on this index, which explains the good adjustment (RMSE 1.39) of the prediction model for the SPAD index [44]. Image-based phenotyping has been successfully employed to detect infestation with plant pathogens, but changes in plant microbiota are still difficult to detect [45]. Our result is particularly interesting since it enables large-scale surveys of croplands inoculated with beneficial microorganisms, allowing detection of positive effects in the plant-microorganism interaction. Moreover, the VARI was strongly associated with chlorophyll content, allowing the efficient and quick selection of croplands inoculated with beneficial microorganisms in an indirect and non-destructible way. This image-based phenotyping method represent an alternative to time-consuming traditional SPAD measurements, enabling the use of images taken using a low-cost remotely piloted aircraft (RPA) equipped with a RGB camera to monitor microbial inoculations.

This research provides evidence of field growth promotion of lettuce by *A. niger* FS1. We demonstrated that even well-nourished plants benefited from *A. niger* inoculation. Therefore, inoculation with *A. niger* surpasses the yield obtained with conventional chemical inputs, allowing productivity gains not reached by just increasing P doses. Moreover, plant phenotyping by traditional measurement of growth parameters and aerial imaging allowed the detection of higher growth and chlorophyll content in inoculated plants. Thus, this research

demonstrates the potential of high-throughput phenotyping methods based on aerial images for large-scale cropland surveys to determine the effect of microorganism inoculation on plant growth.

## Supporting information

**S1 Fig. High-resolution orthoimage showing the aerial view of the experiment.**
(TIF)

## Acknowledgments

The authors thank the Laboratory of Botany/Federal University of Uberlândia for kindly lending the SPAD meter.

## Author Contributions

**Conceptualization:** Patrick Vieira Silva, Gustavo de Souza Marques Mundim, Gabriel Mascarenhas Maciel, Rodrigo Bezerra de Araújo Gallis, Gilberto de Oliveira Mendes.

**Formal analysis:** Patrick Vieira Silva, Lucas Medeiros Pereira, Rodrigo Bezerra de Araújo Gallis, Gilberto de Oliveira Mendes.

**Funding acquisition:** Gilberto de Oliveira Mendes.

**Investigation:** Patrick Vieira Silva, Lucas Medeiros Pereira, Gustavo de Souza Marques Mundim.

**Methodology:** Patrick Vieira Silva, Lucas Medeiros Pereira, Gustavo de Souza Marques Mundim, Gabriel Mascarenhas Maciel, Rodrigo Bezerra de Araújo Gallis, Gilberto de Oliveira Mendes.

**Project administration:** Gilberto de Oliveira Mendes.

**Resources:** Gabriel Mascarenhas Maciel, Gilberto de Oliveira Mendes.

**Supervision:** Gabriel Mascarenhas Maciel, Gilberto de Oliveira Mendes.

**Visualization:** Patrick Vieira Silva, Gilberto de Oliveira Mendes.

**Writing – original draft:** Patrick Vieira Silva.

**Writing – review & editing:** Patrick Vieira Silva, Gabriel Mascarenhas Maciel, Rodrigo Bezerra de Araújo Gallis, Gilberto de Oliveira Mendes.

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
