## [Decision Letter · Decision Letter 0]

26 Aug 2022

PONE-D-22-21385Unlocking lettuce yield potential by Aspergillus niger inoculationPLOS ONE

Dear Dr. Mendes,

Thank you for submitting your manuscript to PLOS ONE. After careful consideration, we feel that it has merit but does not fully meet PLOS ONE’s publication criteria as it currently stands. Therefore, we invite you to submit a revised version of the manuscript that addresses the points raised during the review process.

ACADEMIC EDITOR:Both reviewers have agreed that the title of the manuscript should be improved to reflect the content.Some minor amendments are needed as indicated by both reviewers.

We look forward to receiving your revised manuscript.

Kind regards,

Ali Tan Kee Zuan, Ph.D.

Academic Editor

PLOS ONE

Journal Requirements:

3.Thank you for stating the following financial disclosure: 

"This work was supported by the Fundação de Amparo à Pesquisa do Estado de Minas Gerais (FAPEMIG), grant number APQ-01842-17 to GOM, the Conselho Nacional de Desenvolvimento Científico e Tecnológico (CNPq), grant number 407793/2021-6 to GOM, and NOOA Ciência e Tecnologia Agrícola Ltda, grant to GOM and PVS. "

"I have read the journal's policy and the authors of this manuscript have the following competing interests: a) Research support by the company NOOA Ciência e Tecnologia Agrícola Ltda to PVS and GOM, although the company had no role in the conceptualization, design, data collection, analysis, decision to publish, or preparation of the manuscript; b) Patent application by the Universidade Federal de Uberlândia, Fundação de Amparo à Pesquisa do Estado de Minas Gerais, and NOOA Ciência e Tecnologia Agrícola Ltda; inventors GOM et al.; application number BR 10 2020 000628 2, status: under analysis; specific aspect of manuscript covered in patent application: granular formulation of A. niger used in experiments. All other authors do not have any conflict of interest to declare."

Please respond by return email with your amended Competing Interests Statement and we will change the online submission form on your behalf.

Reviewers' comments:

Reviewer's Responses to Questions

**Comments to the Author**

1. Is the manuscript technically sound, and do the data support the conclusions?

Reviewer #1: Yes

Reviewer #2: Yes

2. Has the statistical analysis been performed appropriately and rigorously? 

Reviewer #1: Yes

Reviewer #2: Yes

3. Have the authors made all data underlying the findings in their manuscript fully available?

Reviewer #1: Yes

Reviewer #2: Yes

4. Is the manuscript presented in an intelligible fashion and written in standard English?

Reviewer #1: Yes

Reviewer #2: Yes

5. Review Comments to the Author

Reviewer #1: In my opinion, the title of this manuscript should be changed to give more focus and emphasis on high-throughput phenotyping methods based on aerial images to determine the effect of microorganism inoculation on plant growth. The technique has potential for large-scale cropland surveys.

Line 141: Sample of the orthoimage in Figure 1 is missing.

Line 263-265: The authors discuss the switch mechanisms of beneficial traits of inoculum and the host plant’s needs. A complete discussion of the mechanisms should be included in the manuscript (discussion section).

Line 302-307: Conclusion should be rewritten with additional emphasis on the potential of aerial imaging techniques to determine the effect of A. niger on growth of lettuce. In addition, the theory about switch mechanisms related to bacterial potency and the need of lettuce (host plants) needs to be explained in depth.

Reviewer #2: Title ambiguous, need to change, not reflected with objective

line 38-39, sentence to rephrase or some grammatical errors

line 46, ...diminish the gap in the crop yield potential.

line 47, to improve sentence.

line 115, by pipetting 3 mL per tray cell 30 min before planting. Need explain, not clear, how do you do this?

line 123-124, explain in detail, approx. size of one granule. how to direct contact with the seedling root?

Figures low resolution, improve it.

6. PLOS authors have the option to publish the peer review history of their article (what does this mean?). If published, this will include your full peer review and any attached files.

Reviewer #1: No

Reviewer #2: No

---

## [Author Response · Author response to Decision Letter 0]

29 Aug 2022

Point-by-point response to comments:

ACADEMIC EDITOR

Both reviewers have agreed that the title of the manuscript should be improved to reflect the content.

Some minor amendments are needed as indicated by both reviewers.

Authors: As suggested by reviewers, we changed the title to better fit the manuscript content. We also corrected all minor issues indicated by reviewers.

Reviewer #1

In my opinion, the title of this manuscript should be changed to give more focus and emphasis on high-throughput phenotyping methods based on aerial images to determine the effect of microorganism inoculation on plant growth. The technique has potential for large-scale cropland surveys.

Authors: We agree with you. We changed the title to give more emphasis on the phenotyping method based on aerial images. 

Line 141: Sample of the orthoimage in Figure 1 is missing.

Authors: Sorry for the mistake. We included Fig 1 and hence renumbered the following figures.

Line 263-265: The authors discuss the switch mechanisms of beneficial traits of inoculum and the host plant’s needs. A complete discussion of the mechanisms should be included in the manuscript (discussion section).

Authors: We improved the discussion at this point, including more details about the mechanisms. See lines 269-273.

Line 302-307: Conclusion should be rewritten with additional emphasis on the potential of aerial imaging techniques to determine the effect of A. niger on growth of lettuce. In addition, the theory about switch mechanisms related to bacterial potency and the need of lettuce (host plants) needs to be explained in depth.

Authors: Conclusive paragraph was rewritten as suggested (lines 311-318). The theory about switching mechanisms was better discussed as well (lines 269-273).

Reviewer #2: 

Title ambiguous, need to change, not reflected with objective

Authors: We changed the title to better fit the manuscript content.

line 38-39, sentence to rephrase or some grammatical errors

Authors: Rephrased. See lines 39-40.

line 46, ...diminish the gap in the crop yield potential.

Authors: Corrected. See line 47.

line 47, to improve sentence.

Authors: Sentence was rephrased. See lines 48-49.

line 115, by pipetting 3 mL per tray cell 30 min before planting. Need explain, not clear, how do you do this?

Authors: Text rephrased for clarity. See lines 115-117.

line 123-124, explain in detail, approx. size of one granule. how to direct contact with the seedling root?

Authors: We included a description of the shape of granules. Also, we have clarified the way we applied the granules to the seedling. See lines 124-126.

Figures low resolution, improve it.

Authors: We have reprocessed all figures using the PACE tool according to the PLOS guidelines.

---

## [Editor Report · Decision Letter 1]

4 Sep 2022

Field evaluation of the effect of Aspergillus niger on lettuce growth using conventional measurements and a high-throughput phenotyping method based on aerial images

PONE-D-22-21385R1

Dear Dr. Mendes,

We’re pleased to inform you that your manuscript has been judged scientifically suitable for publication and will be formally accepted for publication once it meets all outstanding technical requirements.

Kind regards,

Ali Tan Kee Zuan, Ph.D.

Academic Editor

PLOS ONE
---

## [Editor Report · Acceptance letter]

9 Sep 2022

PONE-D-22-21385R1 

Field evaluation of the effect of *Aspergillus niger* on lettuce growth using conventional measurements and a high-throughput phenotyping method based on aerial images 

Dear Dr. Mendes:

I'm pleased to inform you that your manuscript has been deemed suitable for publication in PLOS ONE. Congratulations! Your manuscript is now with our production department. 

Kind regards, 

on behalf of

Dr. Ali Tan Kee Zuan 

Academic Editor

PLOS ONE